# Compound Heterozygous Mutations of *SACS* in a Korean Cohort Study of Charcot-Marie-Tooth Disease Concurrent Cerebellar Ataxia and Spasticity

**DOI:** 10.3390/ijms25126378

**Published:** 2024-06-09

**Authors:** Byung Kwon Pi, Yeon Hak Chung, Hyun Su Kim, Soo Hyun Nam, Ah Jin Lee, Da Eun Nam, Hyung Jun Park, Sang Beom Kim, Ki Wha Chung, Byung-Ok Choi

**Affiliations:** 1Department of Biological Sciences, Kongju National University, Gongju 32588, Republic of Korea; pi0979@naver.com (B.K.P.); jhmom1010@naver.com (A.J.L.); 2Department of Neurology, Korea University Guro Hospital, College of Medicine, Korea University, 148 Gurodong-ro, Guro-gu, Seoul 08308, Republic of Korea; yeonhak59@gmail.com; 3Department of Radiology, Samsung Medical Center, School of Medicine, Sungkyunkwan University, 81 Irwon-ro, Gangnam-gu, Seoul 06351, Republic of Korea; hyunsu83.kim@samsung.com; 4Cell and Gene Therapy Institute, Samsung Medical Center, Gangnam-gu, Seoul 06351, Republic of Korea; naman97@naver.com; 5Department of Domestic Business, Macrogen, Inc., 238 Teheran-ro, Gangnam-gu, Seoul 06221, Republic of Korea; daeun5612@naver.com; 6Department of Neurology, Gangnam Severance Hospital, College of Medicine, Yonsei University, 211 Eonju-ro, Gangnam-gu, Seoul 06273, Republic of Korea; hyungjunpark316@gmail.com; 7Department of Neurology, Kyung Hee University Hospital at Gangdong, College of Medicine, Kyung Hee University, 892 Dongnam-ro, Gangdong-gu, Seoul 05278, Republic of Korea; sbkim@khu.ac.kr; 8Department of Neurology, Samsung Medical Center, School of Medicine, Sungkyunkwan University, 81 Irwonr-ro, Gangnam-gu, Seoul 06351, Republic of Korea; 9Department of Health Science and Technology, Samsung Advanced Institute for Health Sciences and Technology, 81 Irwon-ro, Gangnam-gu, Seoul 06351, Republic of Korea

**Keywords:** autosomal recessive spastic ataxia of Charlevoix-Saguenay disease (ARSACS), cerebellar ataxia, Charcot-Marie-Tooth disease (CMT), Korean, *SACS*

## Abstract

Mutations in the *SACS* gene are associated with autosomal recessive spastic ataxia of Charlevoix-Saguenay disease (ARSACS) or complex clinical phenotypes of Charcot-Marie-Tooth disease (CMT). This study aimed to identify *SACS* mutations in a Korean CMT cohort with cerebellar ataxia and spasticity by whole exome sequencing (WES). As a result, eight pathogenic *SACS* mutations in four families were identified as the underlying causes of these complex phenotypes. The prevalence of CMT families with *SACS* mutations was determined to be 0.3%. All the patients showed sensory, motor, and gait disturbances with increased deep tendon reflexes. Lower limb magnetic resonance imaging (MRI) was performed in four patients and all had fatty replacements. Of note, they all had similar fatty infiltrations between the proximal and distal lower limb muscles, different from the neuromuscular imaging feature in most CMT patients without *SACS* mutations who had distal dominant fatty involvement. Therefore, these findings were considered a characteristic feature in CMT patients with *SACS* mutations. Although further studies with more cases are needed, our results highlight lower extremity MRI findings in CMT patients with *SACS* mutations and broaden the clinical spectrum. We suggest screening for *SACS* in recessive CMT patients with complex phenotypes of ataxia and spasticity.

## 1. Introduction

Charcot-Marie-Tooth disease (CMT), also called hereditary motor and sensory neuropathy (HMSN), is a genetically heterogeneous group of peripheral neuropathic disorders characterized by progressive muscle weakness, atrophy, and sensory loss, primarily in the distal extremities. The genetic underpinnings of CMT have been extensively studied, leading to the identification of more than 130 causative genes [1]. CMT is traditionally considered a rare, simple Mendelian genetic disease but has a loose genotype-phenotype correlation.

CMT is sometimes accompanied by additional symptoms such as central nervous abnormality, myopathy, nephropathy, optic atrophy, hearing loss, ataxia, and spasticity [2], which are believed to occur due to the multi-organ pleiotropic nature of many CMT-relevant genes. *SACS* (MIM 604490), generally known to cause autosomal recessive spastic ataxia of Charlevoix-Saguenay (ARSACS; MIM 270550), is one of the CMT-related genes showing multi-organic complex phenotypes [3,4]. CMT patients with homozygous or compound heterozygous mutations in the *SACS* gene have complex phenotypes, including peripheral polyneuropathy, spinocerebellar ataxia, and spasticity [5,6]. Besides *SACS* mutations, mutations in some genes, including *MFN2*, *REEP1*, *B4GALNT1*, and *C12ORF65*, have been reported to be related with the complex phenotypes of CMT and spasticity [7,8,9].

Since ARSACS was first documented in the 1970s in Quebec [10], linkage analysis mapped its genetic locus to chromosome 13q11 [11]. Then, mutations in *SACS* located within the linked disequilibrium region were identified as the underlying causes of ARSACS [3,4]. Thus far, several hundred homozygous or compound heterozygous mutations have been reported to be the genetic cause of ARSACS [12,13]. In the public databases of ClinVar (https://www.ncbi.nlm.nih.gov/clinvar/ accessed on 3 April 2024) and OMIM (https://www.omim.org/ accessed on 8 May 2024), more than 600 *SACS* variants have been registered as pathogenic or likely pathogenic variants in several recessive disorders, such as spastic ataxia with a pyramidal sign and peripheral neuropathy. Most *SACS* variants occur within short sequences, while several long deletion or insertion variants (copy number variants) have also been reported [14].

The *SACS* gene encodes a SACSIN protein, which integrates with the ubiquitin-proteasome system and HSP70 chaperone machinery [15]. Besides its chaperone activity, the SACSIN protein is believed to have important roles in several cellular processes, including cytoskeletal organization [16,17], mitochondrial homeostasis [18], and cell migration [3,18]. Notably, the regulation of mitochondrial functions by SACSIN may share common mechanisms with many neurodegenerative diseases [16,18,19].

When ARSACS was first reported in Quebec, the affected individuals revealed similar phenotypes characterized by early onset cerebellar ataxia, spasticity, and peripheral neuropathy [3,11]. After that, many cases have shown atypical phenotypes in other ethnic groups. Late-onset patients have been frequently found [20,21], and some Japanese patients showed no spasticity [22,23,24]. ARSACS cases have been reported in East Asian countries, including Japan [22,23,24] and China [14,25,26]. However, limited cases of *SACS* mutations have been reported in Korean patients [27,28,29].

The primary purpose of this study was to identify pathogenic *SACS* mutations in a Korean CMT cohort. As a result, compound heterozygous *SACS* mutations from four CMT families were identified with concurrent ataxia and spasticity. This study also characterized clinical phenotypes, including lower extremity MRI features, and analyzed the genotype-phenotype correlation in the affected individuals with the *SACS* mutations.

## 2. Results

### 2.1. Identification of Compound Heterozygous SACS Mutations

This study performed whole exome sequencing (WES) to identify *SACS* mutations as the genetic causes. As a result, we identified *trans*-arrayed compound heterozygous mutations of *SACS* in four CMT families presenting concurrent cerebellar ataxia and spasticity (Table 1). The eight observed mutations were evaluated to be pathogenic (P) or likely pathogenic (LP) by the American College of Medical Genetics and Genomics and Association for Molecular Pathology (ACMG/AMP) criteria (Appendix A). All the mutations were completely cosegregated with the affected individual(s) in each family with the recessive mode.

In family FC591, c.1966_1967insT (p.S656Ffs*1) and c.4138C>G (p.P1380A) were identified in two affected siblings. Insertion and missense mutations were transmitted from their unaffected father and mother, respectively (Figure 1A). In family FC937, two heterozygous mutations of c.2439_2440delAT (p.V815Gfs*2) and c.10897T>G (p.F3633V) were in a sporadically affected male. The p.F3633V missense mutation was observed in his father; thus, the other 2-bp deletion mutation seemed to be transmitted from his non-examined mother (Figure 1B). The c.2439_2440delAT homozygous mutation has been reported several times in affected individuals with early-onset cerebellar ataxia and peripheral neuropathy from consanguineous families [30,31]. In addition, compound heterozygous *SACS* mutations of c.2439_2440delAT and c.434C>G were observed in a patient with leg spasticity and sensorimotor axonal polyneuropathy [32]. In the FC1157 family, two deletion mutations of c.2903_2906delACAG (p.D968Vfs*13) and c.13217delC (p.T4406Rfs*45) were observed in an affected sister and brother. Unaffected mother had p.T4406Rfs*45, whereas unaffected father showed an incomplete presence of p.D968Vfs*13 at a ratio of 22%, suggesting a gonosomal mosaicism. (Figure 1C). Homozygous mutation of the c.2903_2906delACAG was once reported in a patient with axonal CMT, ataxic gait, and cerebellar phenotype [33]. In the FC1176 family, two stop-gain mutations of c.1596T>A (p.Y532X) and c.3159_3160delCT (p.F1054X) were identified in an affected female. Two mutations were transmitted from each unaffected parent (Figure 1D).

c.2439_2440delAT and c.2903_2906delACAG have been reported as pathogenic mutations in the homozygous or heterozygous states [30,31,32,33]. In contrast, the six other mutations were not reported in public human genome databases, including the International Genome Sample Resource (IGSR), the Korean Reference Genome Database (gnomAD), and the Korean Reference Genome Database (KRGDB). In addition, four pairs of heterozygous mutations found in the analyzed families have not yet been reported as an underlying cause of CMT or ARSACS. Two missense mutations (p.P1380A and p.F3633V) showed very high scores in the genomic evolutionary rate profiling (GERP) with more than 6.0 (Table 2).

Figure 2A,B show sequencing chromatograms and mutation locations on the SACSIN protein, respectively. Two missense mutation sites and their surrounding sequences are highly conserved among vertebrate species, from fish to mammals (Figure 2C). Several in silico analyses by Mutation Taster, REVEL, MUpro, and PolyPhen-2 predicted that all the mutations were pathogenic (Table 2).

### 2.2. Prediction of Conformational Changes by Missense Mutations

Considerable conformational changes were predicted in the mutant proteins due to missense mutations (Figure 3). In the wild protein, the F3633 residue was predicted to connect with the R3636 residue through a cation-π interaction, and the F3666 residue was predicted to have hydrophobic contact with the L3640 and F3633 residues. However, in the p.F3633V mutant, the cation-π interaction with the R3636 residue was replaced by a weak hydrogen bond, and all hydrophobic contacts with the F3666 residue are predicted to be broken. As a result, the distance between surrounding α-helices was predicted to be distant (Figure 3A). In the wild type, the P1380 residue was not connected with the L1360, D1406, D1402, and H1383 residues, but, due to the p.P1380A mutation, a hydrogen bond was predictively created between the P1380A and the H1383 residues. The H1383 residue was linked with the D1406 residue by a hydrogen bond and with the D1402 residue by an ionic interaction. It was also predicted that the distance between the two α-helices will become closer. Additionally, it was predicted that a hydrophobic contact would be created between the D1406 and L1360 residues because the α-helix would be closed. As a result, the distance between the surrounding α-helices was predicted to become closer (Figure 3B). When we applied the PremPS, MAESTROweb, and DynaMut2 programs using the predicted the Protein Data Bank (PDB), it was predicted that the two missense mutations significantly decreased the protein stability (Table 3).

### 2.3. Clinical Manifestation

We examined the clinical features in the affected individuals with *SACS* mutations from the four families (Table 4). The mean age at which neuropathy symptoms first appeared was 9.83 ± 6.46 years (early-to-mid onset ranging from 3 to 17 years), and the age at the time of examination at the hospital was 28.0 ± 4.98 years (22 to 27 years). Table 3 shows the clinical features of six patients with compound heterozygous *SACS* mutations. They showed a similar muscle weakness between the proximal and distal limb muscles. Therefore, their muscle weakness differed from most other CMT patients with length-dependent axonal degeneration. Sensory disturbance was seen in all patients, and the vibration sense was worse than the pain sense. Deep tendon reflexes were hyperactive in all patients. Cerebellar ataxia, spasticity, foot deformity, hearing loss, and nystagmus were observed in all patients. Disability scores showed severe (functional disability scale (FDS) > 3, CMT neuropathy score version 2 (CMTNSv2) > 21) in three patients, and moderate (FDS = 2, 10 < CMTNSv2 ≤ 20) in the other patients.

### 2.4. Electrophysiological Findings

Appendix A shows the results of six patients who underwent nerve conduction studies. These six patients were compatible with HMSN. The upper and lower extremity motor nerve conduction velocities (MNCVs) were slow, and the compound muscle action potentials (CMAPs) were decreased. The sensory nerve damage was more remarkable than the motor nerve damage. In four patients, the upper and lower limb sensory nerve conduction velocities (SNCVs) and sensory nerve action potentials (SNAPs) were not evoked. Hence, the level of sensory neuropathy manifests earlier and exhibits greater severity when contrasted with the degree of motor neuropathy. Brainstem auditory evoked potential (BAEP) was performed on three patients, and all showed a prolonged I-III and III-V interpeak latency, indicating abnormal findings.

### 2.5. MRI Features of the Brain and Lower Extremity

The lower extremity MRI findings for the four patients are summarized in Appendix A. These four patients had minimal to mild fatty infiltration (Goutallier grade 1 or 2) in the lower extremity muscles: II-1 in FC937 (Figure 4A,B), II-2 in FC1157 (Figure 4C,D), II-4 in FC1157 (Figure 4E,F), and II-1 in FC1176 (Figure 4G,H) [34]. Minimal to mild fatty infiltration in the lower extremities was observed in all four patients who underwent MRI. When the lower extremity MRI was performed between 22 and 27 years of age, fatty infiltration was not observed in normal people. Therefore, the fat replacement in these patients proves damage in the lower extremity muscles. Additionally, patients with length-dependent peripheral neuropathies, such as CMT neuropathies, typically have more severe damage to proximal muscles than distal muscles [35,36]. The fact that there was no significant difference in damage to the proximal and distal muscles in these patients is a characteristic finding that differs from other patients with peripheral neuropathy.

Brain MRI scans were performed on three affected individuals, II-2 and II-4 in FC1157 and II-1 in FC1176 (Figure 5). There were no notable findings of significant atrophy or white matter changes in the cerebrum. However, all three individuals showed atrophy, particularly in the superior cerebellar vermis. A bilateral linear hypointensity in the brainstem was observed in the pontine region on the T2 and fluid-attenuated inversion recovery (FLAIR) images.

## 3. Discussion

This is the first cohort study on *SACS* mutations in Korean CMT patients. Thus far, only three sporadic cases have been reported in the Korean population [27,28,29]. In this study, we identified eight heterozygous *SACS* mutations in four recessive early-to-mid onset CMT families with cerebellar spastic ataxia. Among them, six were unreported novel mutations, and the compound heterozygous mutations shown in the examined families were all unreported new combinations of heterozygous mutation pairs. We identified these pairs as the underlying causes of the complex phenotypes observed in polyneuropathy and cerebellar spastic ataxia. This result was based on several key factors: complete cosegregation with affected individuals, genotype-phenotype correlation, unreported or very low frequent mutant alleles, well conservation of mutation sites, in silico prediction of pathogenicity, and likely pathogenic evaluation by the ACMG/AMP criteria.

The prevalence of patients with *SACS* mutations in the Korean CMT cohort was calculated to be 0.3% of the total examined families, while 0.4% of the families were negative for *PMP22* duplication. As a genetic feature, no homozygous *SACS* mutation was found in this study, which is believed to be partly related to the strong prohibition of marriage between relatives. In countries with frequent consanguineous marriages, recessive genetic diseases in many patients have been caused by homozygous mutations [37,38,39], whereas in Korea, recessive conditions in the majority of patients have been reported to be caused by compound heterozygous mutations [40,41].

Mutations in the *SACS* gene are known to affect the expression of the SACSIN protein, leading to mitochondrial dysfunction, influencing the HSP70 chaperone pathway, and ultimately contributing to neurodegeneration. The findings of cerebellar atrophy and bilateral pontine linear hypointensity in this study are commonly observed in other *SACS* mutations. They are associated with cerebellar ataxia and upper motor neuron signs. Other studies have reported the presence of demyelinating or mixed-type polyneuropathy [4,5,26]. However, the cases in this study showed five mixed-type polyneuropathy and one axonal polyneuropathy; thus, no demyelinating polyneuropathy was found.

Sensorimotor polyneuropathy in patients with *SACS* gene have revealed characteristic HMSN phenotypes with other symptoms. In general, it is well known that patients with sensorimotor polyneuropathy show abnormal fatty involvement on lower extremity MRI. Therefore, patients with *SACS* mutations may have lower extremity muscle damage and may be expected to show abnormal fatty infiltration findings on lower extremity MRI. However, lower extremity MRI findings in patients with *SACS* mutations have not been reported. Therefore, this study performed lower extremity MRI on four *SACS* mutation patients with typical clinical features. To our knowledge, this is the first report on lower extremity MRI in patients with *SACS* mutations. All four patients had minimal to mild fatty infiltration in the thigh and calf muscles of the lower extremities. In general, CMT patients show characteristics of length-dependent axonal degeneration; thus, the damage to the calf muscles, the distal part of the lower extremity, is more severe than to the thigh muscles, the proximal part. As the disease gradually worsens, signs of fatty infiltration appear in the proximal thigh muscles. However, in the patients with the *SACS* mutations, the degree of fatty infiltration in the thigh and calf muscles was similar. This was equally observed in all four patients with *SACS* mutations. Thus, these findings require further study with a larger number of *SACS* patients; however, these findings are considered characteristic findings of lower extremity MRI examinations in *SACS* patients.

Most patients in the CMT cohort present with distal dominant impairments, although there are rare cases, such as CMT patients with *TFG* mutations who show distinct proximal dominant involvement [42]. Some ARSACS patients presented with an atypical phenotype, including absence of spasticity and mid-to-late onset [22,23]. Therefore, *SACS* mutations can be considered a genetic cause, if a patient showed similar atrophy between proximal and distal (or thigh and calf) muscles along with only some of the complex phenotypes of cerebellar ataxia, spasticity, and hearing loss.

This study was the first attempt to investigate *SACS* variants thoroughly from a CMT genetic cohort. As a result, biallelic *SACS* variants could be identified as the genetic causes of four recessive CMT families, even though the prevalence was low. We suggest screening of *SACS* variants as the recessive genetic causes of HMSN and cerebellar ataxia in the CMT cohort study.

## 4. Materials and Methods

### 4.1. Patients

This study recruited a cohort consisting of 1889 CMT patients from 1363 unrelated Korean families between April 2005 and May 2023. Then, 413 CMT type 1A (CMT1A) families with the *PMP22* duplication were excluded. Thus, the remaining 950 families were screened for *SACS* mutations. Written informed consent was obtained from all participants by a protocol approved by the Institutional Review Boards of Sungkyunkwan University, Samsung Medical Center (2014-08-057-002) and Kongju National University (KNU_IRB_2018-62). Parents provided written consent for the minors involved in this study.

### 4.2. Molecular Genetic Analysis

Genomic DNA was purified from whole blood using the QIAamp DNA Mini Kit (Qiagen, Hilden, Germany). The affected individuals, who tested negative for *PMP22* duplication, underwent WES. The exomes were captured using the SureSelect Human All Exon 50M Kit (Agilent Technologies, Santa Clara, CA, USA), and sequencing was performed using the HiSeq 2500 Genome Analyzer (Illumina, San Diego, CA, USA).

The human genome assembly hg19 (GRCh37) was used as the reference sequence (http://genome.ucsc.edu/ accessed on 14 January 2024). VCF files were generated from the FastQ files of the WES low data using the GATK and SAMtools algorithms. Functionally significant variants (missense, nonsense, exonic indel, and splicing site variants) were first selected from the VCF files. Then, unreported or rare variants with allele frequencies of ≤ 0.01 were subsequently chosen from CMT- and spastic ataxia-related genes. Minor allele frequencies (MAF) were cross-referenced with various public databases, including dbSNP (http://www.ncbi.nlm.nih.gov/ accessed on 15 January 2024), IGSR (https://www.internationalgenome.org/ accessed on 8 April 2024), gnomeAD (https://gnomad.broadinstitute.org/ accessed on 8 April 2024), and KRGDB (http://coda.nih.go.kr/coda/KRGDB/ accessed on 8 April 2024). The identified variants were confirmed through Sanger sequencing using the SeqStudio Genetic Analyzer (Life Technologies-Thermo Fisher Scientific, Foster City, CA, USA).

### 4.3. Conservation, Conformational Change, and In Silico Prediction of Mutant Proteins

Conservation analysis of the mutation sites was conducted using MEGA6, ver. 6.0 (http://www.megasoftware.net/ accessed on 13 March 2024). GERP scores were determined using the GERP++ program (http://mendel.stanford.edu/SidowLab/downloads/gerp/). In silico prediction of mutations was performed using the Mutation taster (https://www.genecascade.org/MutationTaster2021/ accessed on 26 March 2024), MUpro (http://www.ics.uci.edu/~baldig/mutation/ accessed on 26 March 2024), PolyPhen-2 (http://genetics.bwh.harvard.edu/pph2/ accessed on 26 March 2024), and REVEL (https://sites.google.com/site/revelgenomics/ accessed on 26 March 2024). Pathogenicity was evaluated using the guidelines established by the ACMG/AMP criteria (http://wintervar.wglab.org/ accessed on 4 April 2024). Three-dimensional conformational changes of mutant proteins were predicted using I-TASSER (https://seq2fun.dcmb.med.umich.edu//I-TASSER/ accessed on 25 April 2024) and were visualized through the Mol*features of the Protein Data Bank (https://www.rcsb.org/3d-view/ accessed on 25 April 2024). Based on the generation of PDB, the effects of the mutations on the protein stability were evaluated using the MAESTROweb (https://biwww.che.sbg.ac.at/maestro/web/ accessed on 25 April 2024), PremPS (https://lilab.jysw.suda.edu.cn/research/PremPS/ accessed on 25 April 2024), and DynaMut2 (http://biosig.unimelb.edu.au/dynamut2/ accessed on 25 April 2024) programs.

### 4.4. Clinical Assessment

Clinical data were collected in a standardized manner, including evaluations of motor and sensory impairments, deep tendon reflexes, and muscle atrophy. The strength of the flexor and extensor muscles was assessed manually using the widely accepted Medical Research Council (MRC) scale [43]. CMTNSv2 and FDS were used to quantify physical disability. Sensory impairments were evaluated based on the level and severity of pain, temperature perception, vibration sense, and positional awareness. The ages of onset were determined through patient reports, specifically by inquiring about the timing and age at which initial symptoms, such as distal muscle weakness, foot deformities, or sensory changes, first manifested.

### 4.5. Electrophysiological Examination

Motor and sensory conduction velocities of the median, ulnar, peroneal, tibial, and sural nerves were assessed using the standard methods of surface stimulation and recording electrodes. MNCVs and CMAPs of the median and ulnar nerves were measured by stimulating at both the elbow and wrist. Similarly, MNCVs and CMAPs of the peroneal and tibial nerves were determined by stimulation at the knee and ankle. CMAPs were quantified by measuring the difference between the baseline and negative peak values. SNCVs were evaluated over a finger-wrist segment for the median and ulnar nerves using orthodromic scoring, and recordings were also obtained for the sural nerves. SNAPs were determined by measuring the amplitude between the positive and negative peaks. BAEP was obtained by monitoring the two channels placed in the bilateral mastoid processes (A1 and A2, left and right, respectively) referenced to the Cz position when 43.9 Hz auditory stimuli were given by 1.5 ms/div using an in-ear microphone. Bilateral latencies and amplitudes from wave I-V were recorded at baseline.

### 4.6. Brain and Lower Extremity MRI

MRI scans were conducted in a supine position using the Ingenia 3.0T CX MRI system (Philips Healthcare, Best, Netherlands). T1-weighted axial and sagittal images and T2-weighted axial FLAIR and susceptibility weighted images (SWIs) were sequentially obtained to evaluate the brain involvement. Comprehensive MRI scans were obtained in the pelvic girdle, bilateral thigh, and lower legs for lower extremity images. Fatty infiltration in the thigh and lower leg muscles was assessed on axial T1-weighted turbo spin-echo images using a five-point semiquantitative scale [34]. Thigh and lower leg muscles were evaluated bilaterally at two levels (proximal and distal), respectively.

## 5. Conclusions

This study identified eight *SACS* mutations in four recessive CMT families as the underlying cause of the complex phenotypes. Six were unreported novel mutations, and the compound heterozygous mutations shown in the examined families were all unreported new combinations of mutation pairs. The prevalence of CMT families with *SACS* mutations was determined to be 0.3% in the Korean CMT cohort. This study provides new insights into *SACS* mutations in a Korean CMT cohort study, revealing their genetic patterns, prevalence, and clinical and neuroimaging manifestations. All the patients showed similar fatty infiltrations between the proximal and distal lower limb muscles, which differed from the CMT patients without *SACS* mutations who had distal dominant fatty involvement. The consistent clinical presentation aligns with established knowledge, and the distinctive lower extremity MRI findings broaden the clinical spectrum of CMT patients with *SACS* mutations. This study emphasizes the testing of the *SACS* gene for recessive CMT patients showing complex symptoms of cerebellar ataxia and spasticity.

## Figures and Tables

**Figure 1 ijms-25-06378-f001:**
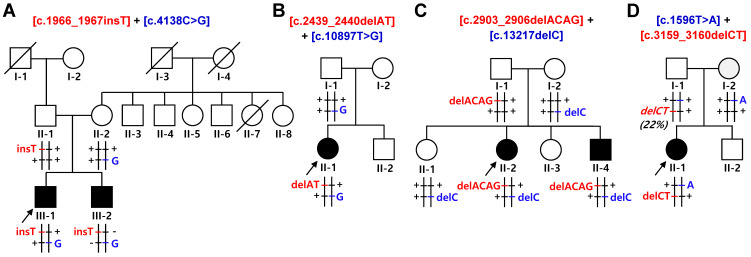
Pedigrees of autosomal recessive CMT families with spastic ataxia. Compound heterozygous mutations in *SACS* were found in all the affected individuals. Haplotypes of two mutation pairs in each family are indicated at the bottom of all the examined individuals with brief chromosomal signs. Black and white symbols represent affected and unaffected individuals, respectively. The probands are indicated by an arrow (□: male, and ○: female). Mutations transmitted from the father and mother are shown in red and blue, respectively, and common (wild) alleles are indicated by “+”. Mutant allele and percentage in parentheses with italic at the bottom of the I-1 in (**D**) represent a gonosomal mosaicism and its ratio. (**A**) FC591, (**B**) FC937, (**C**) FC1157, and (**D**) FC1176.

**Figure 2 ijms-25-06378-f002:**
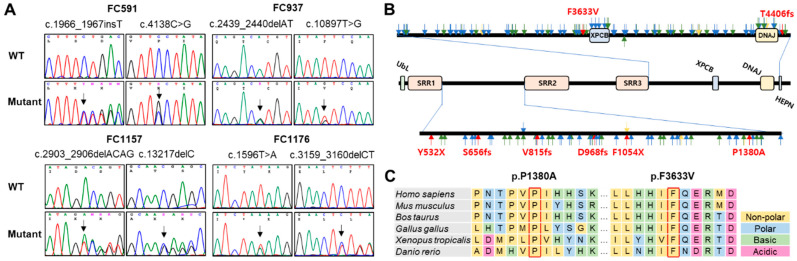
Identification and conservation of *SACS* mutations. (**A**) Sequencing chromatograms of the *SACS* mutations. The mutation sites are indicated by arrows. (**B**) Schematic structure of SACSIN protein. Mutations identified in this study are indicated by red arrows. Blue, green, and pink arrows indicate previously reported frameshift, nonsense, and missense mutations, respectively (DNAJ: DnaJ molecular chaperone homology domain; HEPN: higher eukaryotes and prokaryotes nucleotide-binding domain; SRR: SACSIN repeating region; UbL: ubiquitin-Like domain; XPCB: xeroderma pigmentosum complementation group C binding domain). (**C**) Conservation of missense mutation sites. Two missense mutations are located near SRR2 (p.P1380A) and XPCB (p.F3633V), and their sites were well conserved among vertebrate species from fish to mammals.

**Figure 3 ijms-25-06378-f003:**
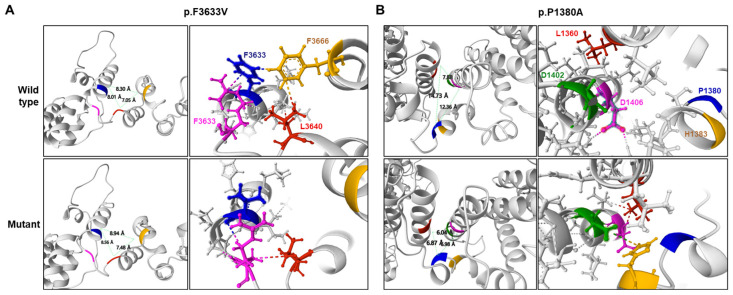
Prediction of 3D structural changes due to missense mutations. Mutated residues are shown in blue, and surrounding residues predicted to undergo structural changes due to the mutations are shown in yellow, pink, red, and green. (**A**) p.F3633V, (**B**) p.P1380A.

**Figure 4 ijms-25-06378-f004:**
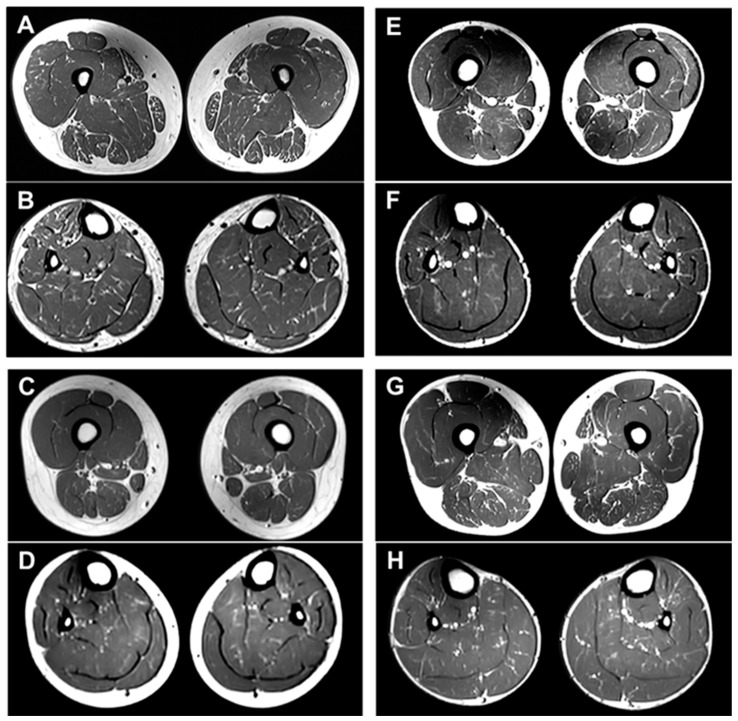
T1-weighted lower limb MR images of four patients with *SACS* mutations: II-1 (F/27 years) in FC937 (**A**,**B**), II-2 (F/26 years) in FC1157 (**C**,**D**), II-4 (M/25 years) in FC1157 (**E**,**F**), and II-1 (M/22 years) in FC1176 (**G**,**H**). All four patients had minimal to mild fatty infiltration in the thigh and calf muscles of the lower extremities, and there was no significant difference in the degree of fatty replacements between proximal and distal muscles.

**Figure 5 ijms-25-06378-f005:**
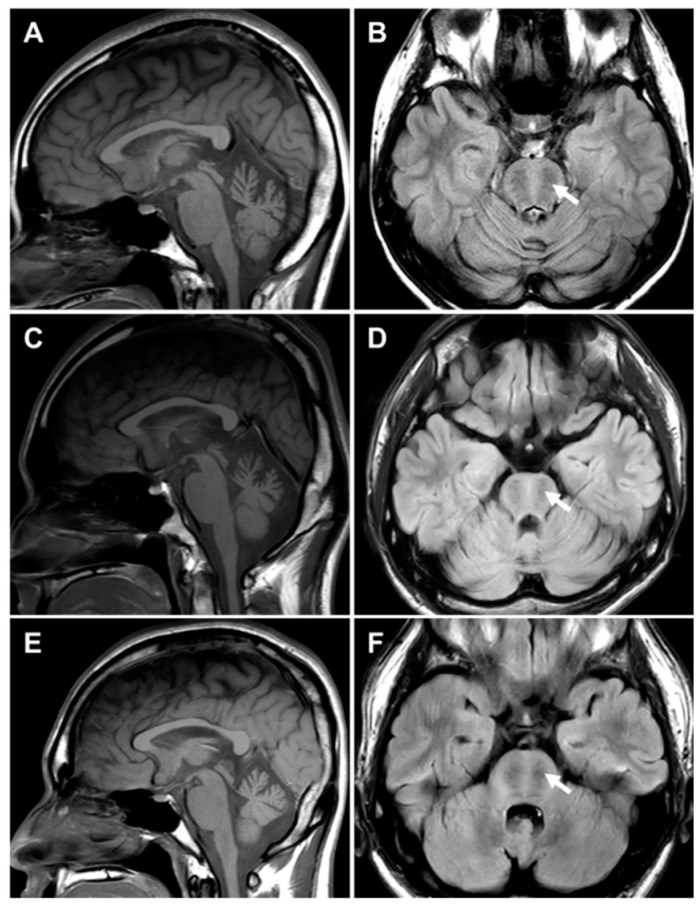
Brain MRIs of three CMT patients with *SACS* mutations. Brain images showed cerebellar atrophy, especially in the superior cerebellar vermis. Bilateral linear hypointense striations were observed in the pontine region (arrows). (**A**,**B**) II-2 (F/26 years) in FC1157, (**C**,**D**) II-4 (M/25 years) in FC1157, and II-1 (M/22 years) in FC1176 (**E**,**F**).

**Table 1 ijms-25-06378-t001:** Compound heterozygous mutations of *SACS* in CMT patients with cerebellar ataxia.

Family ID	Mutations		Clinical Phenotype	ACMG/AMP
	Nucleotide ^1^	Amino Acid ^1^		
FC591	[c.1966_1967insT] + [c.4138C>G]	[p.S656Ffs*1] + [p.P1380A]	CMT, cerebellar ataxia, spasticity, and HL	P p
FC937	[c.2439_2440delAT] + [c.10897T>G]	[p.V815Gfs*2] + [p.F3633V]	CMT, cerebellar ataxia, and spasticity, HL	P LP
FC1157	[c.2903_2906delACAG] + [c.13217delC]	[p.D968Vfs*13] + [p.T4406Rfs*45]	CMT, cerebellar ataxia, spasticity, and HL	P P
FC1176	[c.1596T>A] + [c.3159_3160delCT]	[p.Y532X] + [p.F1054X]	CMT, cerebellar ataxia, spasticity, and HL	P P

^1^ Reference sequences of nucleotides: NM_014363.6 and amino acids: NP_055178.3.

**Table 2 ijms-25-06378-t002:** Allele frequencies and in silico prediction of the *SACS* variants.

Variants ^1^	dbSNP Acc. No.	Mutant Allele Frequencies			GERP	In Silico Analyses ^2^				References
		IGSR	gnomAD	KRGDB		MutT	REVEL	PP2	MU	
p.Y532X	rs2137720760	-	-	-	−2.77	200/0 *	-	-	-	
p.S656Ffs*1	-	-	-	-	5.14	200/0 *	-	-	-	
p.V815Gfs*2	rs775059063	-	0.0000142	-	−1.5	200/0 *	-	-	-	[30,31,32]
p.D968Vfs*13	rs1259615333	-	0.0000099	-	3.71	199/1 *	-	-	-	[33]
p.F1054X	rs2137637877	-	-	-	5.09	200/0 *	-	-	-	
p.P1380A	-	-	-	-	6.06	66/34 *	0.637 *	0.965 *	−0.757 *	
p.F3633V	rs1382541188	-	0.000013	-	6.04	72/28 *	0.626 *	0.999 *	−0.292 *	
p.T4406Rfs*45	-	-	-	-	5.85	198/2 *	-	-	-	

^1^ Reference amino acid sequence: NP_055178.3. ^2^ In silico scores of Mutation taster (MutT) > 0.5, Rare Exome Variant Ensemble Learner (REVEL) > 0.5, PolyPhen-2 (PP2) ~1, and MUpro (MU) < 0 indicate pathogenic prediction (* denotes a pathogenic prediction).

**Table 3 ijms-25-06378-t003:** Prediction of protein stabilities by the missense mutations.

Mutation	Protein Stability Prediction Tools ^1^		
	PremPS	MAESTROweb	DynaMut2
p.F3633V	0.68 *	0.283 *	−0.73 *
p.P1380A	0.91 *	0.121 *	−1.10 *

^1^ Protein stability scores of PremPS > 0, MAESTROweb > 0, and DynaMut2 < 0 indicate instability mutations (* denotes an instability prediction).

**Table 4 ijms-25-06378-t004:** Clinical manifestations of the CMT patients with *SACS* mutations.

Family ID	FC591		FC937	FC1157		FC1176
Patients	III-1	III-2	II-1	II-2	II-4	II-1
Mutations	p.S656Ffs*1 + p.P1380A		p.V815Gfs*2 + p.F3633V	p.D968Vfs*13 + p.T4406Rfs*45		p.Y532X + p.F1054X
Sex	Male	Male	Female	Female	Man	Man
Examined age (years)	35	33	27	26	25	22
Onset age (years)	4	5	15	17	15	3
Muscle weakness						
Upper limb (MRC)						
Proximal (Right/Left) ^1^	4+/4+	4+/4+	4/4	4+/4+	4+/4+	4/4
Distal (Right/Left) ^2^	4+/4+	4+/4+	4/4	4+/4+	4+/4+	4/4
Lower limb (MRC)						
Proximal (Right/Left) ^3^	4/4	4+/4+	4-/4-	4+/4+	4/4	4/4
Distal (Right/Left) ^4^	4/4	4+/4+	4-/4-	4+/4+	4/4	4/4
Muscle atrophy	Mild	Minimal	Moderate	Minimal	Mild	Mild
Sensory disturbance	Yes	Yes	Yes	Yes	Yes	Yes
DTR ^5^						
Biceps jerk reflex	++	++	+++	+++	+++	+++
Knee jerk reflex	+++	+++	+++	+++	+++	+++
Disability score						
FDS	2	2	4	2	4	4
CMTNSv2	13	11	24	17	22	23
Pyramidal sign	Yes	Yes	Yes	Yes	Yes	Yes
Spasticity	Yes	Yes	Yes	Yes	Yes	Yes
Cerebellar ataxia	Yes	Yes	Yes	Yes	Yes	Yes
Dysarthria	No	No	Yes	Yes	No	Yes
Nystagmus	Yes	Yes	Yes	Yes	Yes	Yes
Foot deformity	Yes	Yes	Yes	Yes	Yes	Yes
Intellectual disability	No	No	No	No	No	No
Hearing loss	Yes	Yes	Yes	Yes	Yes	Yes
Electrophysiology						
Nerve conduction	HMSN	HMSN	HMSN	HMSN	HMSN	HMSN
BAEP	NA	NA	Abnormal	Abnormal	NA	Abnormal
Brain MRI	NA	NA	NA	Cerebellar atrophy	Cerebellar atrophy	Cerebellar atrophy

^1^ Proximal (upper limb): elbow flexion MRC grade, ^2^ distal (upper limb): finger abduction MRC grade, ^3^ proximal (lower limb): knee flexion MRC grade, ^4^ distal (lower limb): ankle dorsiflexion MRC grade. ^5^ DTR: ++: normal reflex, +++: increase reflex.

## Data Availability

Data available on request from the authors.

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
