# Peer review of "Compound Heterozygous Mutations of SACS in a Korean Cohort Study of Charcot-Marie-Tooth Disease Concurrent Cerebellar Ataxia and Spasticity"

_ijms, 2024, doi:10.3390/ijms25126378_

Round 1

Reviewer 1 Report

Comments and Suggestions for Authors

This is a very good article for this rare condition. 

This is comprehensive description of this rare type of CMT. The table and figure clearly explain the features. English is excellent and narrative is smooth. Authors are trying to emphasize the MRI features which apparently are minimal or no fatty infiltration. I would suggest that authors rephrase this aspect of description.

I have the following comments for review:

·         Abstract:

 ‘’Lower limb magnetic resonance imaging (MRI) was done in four patients, and all had fatty replacements. Of note, they all had similar fatty infiltrations between the proximal and distal lower limb muscles, different from the neuromuscular imaging feature in most CMT patients without SACS mutations who had distal dominant fatty involvement.

Therefore, these findings were considered a characteristic feature in CMT patients with SACS mutations.’’

This segment raises the expectation to find specific MRI features which apparently are missing in Figure 4

·         Figure 4:

‘’MRIs were prepared at the thigh level (A, C, E, G) or calf level (B, D, F, H).’’

 Please lay out the picture in the format that is easy to follow. I would suggest both proximal and distal MRI of each patient in same row.

Lines: 219-220

‘’Notably, no clear predominance of fatty changes was observed at the distal levels or in specific

compartment muscles.’’

This is contradicting your lines 223- 227

·         225-227

‘’Of note, there were no cases in which normal muscle exists without fatty infiltration. Considering that the age 226 at which the MRI was taken was from 22 to 27 years old, the fatty replacement on the 227 lower extremity MRI was very unusual.

Do you mean that the MRI was normal? In that case rephrase it so that it is easy for reader to understand.

You need to put reference of studies for CMT and MRI here for ‘’ Of note, there were no cases in which normal muscle exists without fatty infiltration.’’ (line 225)

·     

    267-271

‘’Therefore, patients with SACS mutations may have lower extremity muscle damage and may be expected to show abnormal fatty infiltration findings on lower extremity MRI. However, lower extremity MRI findings in patients with SACS mutations have not been reported. Therefore, this study performed lower extremity MRI on four SACS mutation patients with typical clinical features.’’

Authors have made a strong assumption here that severe pathology is associated with significant fatty infiltration of muscles which is not the case for many neuromuscular conditions. It is still good to report normal or minimal changes.

Author Response

This is a very good article for this rare condition. This is comprehensive description of this rare type of CMT. The table and figure clearly explain the features. English is excellent and narrative is smooth. Authors are trying to emphasize the MRI features which apparently are minimal or no fatty infiltration. I would suggest that authors rephrase this aspect of description.

I have the following comments for review:

Abstract: “Lower limb magnetic resonance imaging (MRI) was done in four patients, and all had fatty replacements. Of note, they all had similar fatty infiltrations between the proximal and distal lower limb muscles, different from the neuromuscular imaging feature in most CMT patients without SACS mutations who had distal dominant fatty involvement. Therefore, these findings were considered a characteristic feature in CMT patients with SACS mutations.” This segment raises the expectation to find specific MRI features which apparently are missing in Figure 4.

Figure 4: “MRIs were prepared at the thigh level (A, C, E, G) or calf level (B, D, F, H).” Please lay out the picture in the format that is easy to follow. I would suggest both proximal and distal MRI of each patient in same row.

[Answer]

Thanks for the advice. As you advised, the MRI images were laid out in an easy-to-follow format (see Figure 4). Each patient's proximal (thigh level) and distal (calf level) lower extremity MRIs were placed in the same column. In addition, the Figure 4 legend was also revised according to the comments.

Lines: 219-220: “Notably, no clear predominance of fatty changes was observed at the distal levels or in specific compartment muscles.” This is contradicting your lines 223- 227

[Answer]

Thank you very much for the advice. But, we think this sentence does not contradict the previous statement. In the common typical CMT patients, there are notable differences on lower extremity MRI. The anterior or lateral compartment muscles typically being more severely damaged than the posterior compartment muscles at the calf level. However, no clear predominance of fatty changes in the muscles of these compartments was observed in the patients with SACS. These are the characteristics of SACS patients. As your advice, the corresponding sentence was changed as follows: All four patients had minimal to mild fatty infiltration in the thigh and calf muscles of the lower extremities, and there was no significant difference in the degree of fatty replacements between proximal and distal muscles.

225-227: ‘’Of note, there were no cases in which normal muscle exists without fatty infiltration. Considering that the age at which the MRI was taken was from 22 to 27 years old, the fatty replacement on the lower extremity MRI was very unusual. Do you mean that the MRI was normal? In that case rephrase it so that it is easy for reader to understand.

You need to put reference of studies for CMT and MRI here for “Of note, there were no cases in which normal muscle exists without fatty infiltration.”(line 225)

[Answer]

In normal people, if a lower extremity MRI is done between the ages of 22 and 27 years, the lower extremity muscles are normal and there is no fat replacement. Therefore, it was very unusual to observe fatty replacements on lower extremity MRI. As you advised, I modified the entire paragraph, including the sentences you pointed out, to make it easier to understand:

From: Of note, there were no cases ……. e severe damage to the distal part in the other CMT subtypes.

To: Minimal to mild fatty infiltration was observed in all four patients who underwent MRI. When the lower extremity MRI was performed between 22 and 27 years of age, fatty infiltration was not observed at all in normal people. Therefore, the observation of fat replacement on lower extremity MRI in SACS patients aged 22 to 27 years proves that there was damage to the lower extremity muscles. Additionally, patients with length-dependent peripheral neuropathies, such as CMT neuropathies, typically have more severe damage to proximal muscles than distal muscles [35,36] Considering these differences, the fact that there was no significant difference in the degree of damage to the proximal and distal muscles in SACS patients was a characteristic finding that was different from other peripheral neuropathy patients.

As the comment, we also added two references, [35] Kim et al (2021) and [36] Kwon et al (2021) in the following sentence: Additionally, patients with length-dependent peripheral neuropathies, such as CMT neuropathies, typically have more severe damage to proximal muscles than distal muscles [35,36].

267-271: ‘’Therefore, patients with SACS mutations may have lower extremity muscle damage and may be expected to show abnormal fatty infiltration findings on lower extremity MRI. However, lower extremity MRI findings in patients with SACS mutations have not been reported. Therefore, this study performed lower extremity MRI on four SACS mutation patients with typical clinical features.’’

Authors have made a strong assumption here that severe pathology is associated with significant fatty infiltration of muscles which is not the case for many neuromuscular conditions. It is still good to report normal or minimal changes.

[Answer]

Thanks for the good advice. I agreed with the opinion that it is better to report minimal to mild changes, so we revised it as follows. And as you advised, we modified the entire paragraph including the sentences you pointed out.

From: Sensorimotor polyneuropathy in patients with SACS gene mutations is well known. ……… Thus, these findings require further study with a larger number of SACS patients; however, these findings are considered characteristic findings of lower extremity MRI examinations in SACS patients.

To: Sensorimotor polyneuropathy in patients with SACS gene mutations is well known. Additionally, it is well known that patients with sensorimotor polyneuropathy show abnormal fatty involvement on lower extremity MRI. Therefore, it can be easily expected that patients with SACS mutations have lower extremity muscle damage and abnormal fatty infiltration on their lower extremity MRI. However, lower extremity MRI findings in patients with SACS mutations have not been reported. Therefore, in this study, lower extremity MRI was performed on four SACS mutation patients with typical clinical features. To our knowledge, this is the first report on lower extremity MRI in patients with SACS mutations. All four patients had minimal to mild fatty infiltration in the thigh and calf muscles of the lower extremities. Patients with peripheral neuropathy, including CMT, exhibit features of length-dependent axonal degeneration. Therefore, damage to the calf muscle, which is the distal part of the lower extremity, is more serious damage than to the thigh muscle, which is the proximal part. And as the disease gradually worsens, signs of fatty infiltration appear in the proximal thigh muscles. However, in patients with SACS mutations, the extent of fatty infiltration in the thigh and calf muscles was similar. This was equally observed in all four patients with SACS mutations. Therefore, these findings are considered characteristic features on lower extremity MRI in SACS patients, although further studies with a larger number of SACS patients are needed.

Reviewer 2 Report

Comments and Suggestions for Authors

The authors presented an interesting and original manuscript related to the sacsinopathies, especially the context of SACS-related CMT. The group of cases presented by the authors in the manuscript enabled the identification of patients with relative suggestive features: early-onset of disease in the first two decades of life, axonal/mixed sensorimotor polyneuropathy (predominantly sensory), retained or brisk reflexes, pyramidal signs, hearing loss, spastic-ataxia, pes cavus and typical neuroimaging features suggesting sacsinopathy (such as in ARSACS). Figure 4 showing muscle MR imaging studies is extremely original and contributes markedly to the current knowledge about this group of disorders. Some aspects could be reviewed by the authors at this point to increase the quality of their content: 

1. I have an important suggestion that in this manuscript authors should change the order in which they have presented their text. Results are in Part 2; Materials and Methods are in Part 4. This order does not make any sense, even in a manuscript focused in aspects associated with clinical aspects. So, I suggest authors to present their content in a classical way: Introduction; Materials and Methods; Results; Discussion; Conclusion. 

2. All human genes that have been cited in the text should be presented in italics (lines 58, 60 and 61). 

3. I suggest authors to include their neuroimaging figure showing typical features seen in ARSACS (which is currently in supplementary file) in the manuscript content. 

4. Discussion of the manuscript is short. I suggest authors to include a brief paragraph describing the clinical and neurophysiological summary of the phenotype seen in this type of CMT and comparing with other differential diagnosis related to other genetic subtypes of CMT (i.e., MFN2 variants) or other neurological conditions (i.e., metabolic diseases). 

Author Response

The authors presented an interesting and original manuscript related to the sacsinopathies, especially the context of SACS-related CMT. The group of cases presented by the authors in the manuscript enabled the identification of patients with relative suggestive features: early-onset of disease in the first two decades of life, axonal/mixed sensorimotor polyneuropathy (predominantly sensory), retained or brisk reflexes, pyramidal signs, hearing loss, spastic-ataxia, pes cavus and typical neuroimaging features suggesting sacsinopathy (such as in ARSACS). Figure 4 showing muscle MR imaging studies is extremely original and contributes markedly to the current knowledge about this group of disorders. Some aspects could be reviewed by the authors at this point to increase the quality of their content:

  1. I have an important suggestion that in this manuscript authors should change the order in which they have presented their text. Results are in Part 2; Materials and Methods are in Part 4. This order does not make any sense, even in a manuscript focused in aspects associated with clinical aspects. So, I suggest authors to present their content in a classical way: Introduction; Materials and Methods; Results; Discussion; Conclusion.

[Answer]

Thank you very much for your comment. You suggested a content order in a classical way, but we should follow the template provided by the journal. So we didn't change the content order. However, several modifications were made, including changing the positions of full words and abbreviations of several terms.

  1. All human genes that have been cited in the text should be presented in italics (lines 58, 60 and 61).

[Answer]

Thank you very much for your comment. We checked gene names through the manuscript and gene names (SACS, MFN2, REEP1, B4GALNT1, and C12ORF65) were indicated by italic.

  1. I suggest authors to include their neuroimaging figure showing typical features seen in ARSACS (which is currently in supplementary file) in the manuscript content.

[Answer]

Thank you very much for your comment. Figure S1 (Brain MRIs of three CMT patients with SACS mutations) were transferred to Figure 5 as your comment.

  1. Discussion of the manuscript is short. I suggest authors to include a brief paragraph describing the clinical and neurophysiological summary of the phenotype seen in this type of CMT and comparing with other differential diagnosis related to other genetic subtypes of CMT (i.e., MFN2 variants) or other neurological conditions (i.e., metabolic diseases).

[Answer]

Thank you very much for your comment. As the comment, we added a paragraph at the top of the last paragraph of Discussion: Most patients in the CMT cohort present with distal dominant impairments, although there are rare cases, such as CMT patients with TFG mutations who show distinct proximal dominant involvement [42]. Some ARSACS patients presented with an atypical phenotype, including absence of spasticity and mid-to-late onset [22,23]. Therefore, SACS mutations can be considered a genetic cause, if a patient showed similar atrophy between proximal and distal (or thigh and calf) muscles along with only some of the complex phenotypes of cerebellar ataxia, spasticity, and hearing loss.